# Structures of the human peroxisomal fatty acid transporter ABCD1 in a lipid environment

Le Thi My Le [1,2], James Robert Thompson [1,2], Phuoc Xuan Dang[1], Janarjan Bhandari [1] & Amer Alam [1✉]

The peroxisomal very long chain fatty acid (VLCFA) transporter ABCD1 is central to fatty acid catabolism and lipid biosynthesis. Its dysfunction underlies toxic cytosolic accumulation of VLCFAs, progressive demyelination, and neurological impairments including X-linked adrenoleukodystrophy (X-ALD). We present cryo-EM structures of ABCD1 in phospholipid nanodiscs in a nucleotide bound conformation open to the peroxisomal lumen and an inward facing conformation open to the cytosol at up to 3.5 Å resolution, revealing details of its transmembrane cavity and ATP dependent conformational spectrum. We identify features distinguishing ABCD1 from its closest homologs and show that coenzyme A (CoA) esters of VLCFAs modulate ABCD1 activity in a species dependent manner. Our data suggest a transport mechanism where the CoA moieties of VLCFA-CoAs enter the hydrophilic transmembrane domain while the acyl chains extend out into the surrounding membrane bilayer. The structures help rationalize disease causing mutations and may aid ABCD1 targeted structure-based drug design.

[1] The Hormel Institute, University of Minnesota, Austin, MN 55912, USA. [2] These authors contributed equally: Le Thi My Le, James Robert Thompson. ✉email: aalam@umn.edu

X-ALD (OMIM#300100) is the most common inherited peroxisomal disorder, with a prevalence of approximately 1/20,000 births. Childhood-onset cerebral adrenoleukodystrophy (CCALD) and adult-onset adrenomyeloneuropathy (AMN) are two main forms of X-ALD[1]. A feature of X-ALD is a build-up of high levels of VLCFAs containing 24 or more carbons throughout the body, which can cause damage to the nervous system due to progressive demyelination[2]. While X-ALD presents as a metabolic neurodegenerative disorder, phenotypic variability is high[3]. Dysfunction of the peroxisomal VLCFA transporter ABCD1/ALDP is identified as an underlying cause of X-ALD, with over 800 disease-causing mutations of the *ABCD1* gene identified[4–7].

ABCD1 is a peroxisome-membrane spanning protein that mediates the import of various VLCFAs into the peroxisome in an ATP hydrolysis-dependent manner[8–12] for peroxisome specific beta-oxidation. Accordingly, ABCD1 functional deficiency impairs the degradation of VLCFAs[13–16] and may also alter distributions of phospholipid and lysophospholipid species in different brain regions[17]. ABCD1 is related to and displays functional overlap with two other peroxisomal transporters: ABCD2, also known as ALD related protein/ALDRP[18], and ABCD3, also known as peroxisome membrane protein 70/PMP70[19,20], with which it shares 62 and 39% sequence identity, respectively. These three proteins all transport acyl-CoA substrates, and despite functional overlap, display distinct substrate specificities[9,21,22]. However, the structural requirements for differing specificity remain unknown, and only mutations in ABCD1 are associated with X-ALD. A fourth ABCD family member, ABCD4 (24% sequence identity to ABCD1), is a lysosome-specific transporter that transports vitamin B12[23–25]. Unlike its peroxisomal counterparts as well as canonical ABC exporters it shares a fold with, ABCD4 reportedly functions as an importer, with substrate (Vitamin B12/Cobalamin) suggested to bind the ATP bound 'outward' facing conformation open to the lysosomal lumen and its cytoplasmic release occurring from an 'inward open' conformation[26]. All four ABCD proteins are half-transporters expressed as single polypeptides containing a transmembrane domain (TMD) and a nucleotide-binding domain (NBD) that must dimerize to form the functional transporter. While some evidence is reported for the existence of ABCD heterodimers[11,27,28], functional homodimers appear to be most prevalent in vivo[21,29].

Structural insights into ABCD1 are currently limited to those gleaned from homology models based on related transporters[30] and, more recently, the cryo-EM structure of nucleotide bound ABCD4 in detergent[26]. To define the ABCD1 structural elements that enable ATP dependent and specific VLCFA transport at the molecular level, we determined its structures in nanodisc reconstituted form in both a nucleotide-bound outward open (OO) conformation open to the peroxisomal lumen, and an inward open (IO) conformation open to the cytosol. In conjunction with ATP-driven functional assessments, our data reveal the conformational landscape associated with the ATP-dependent transport cycle of ABCD1 and highlight key features of its TMD that offer insights into its potential substrate-binding mechanism. These structures also allow us to pinpoint the location of the most frequently occurring disease-causing mutations in the ABCD1 TMD that will allow for the conceptualization of structure-function hypotheses based on X-ALD patient mutations[31]. Finally, these structures open the door for more accurate structure-guided design of ABCD1 targeted small molecule therapeutics and computational studies of ABCD1 structure and function.

## Results

**In vitro ATPase activity of ABCD1 is modulated by VLCFA-CoAs in a species-dependent manner**. We utilized a tetracycline-inducible stable cell line to produce human ABCD1 in human embryonic kidney (HEK) 293 cells and tested its ATPase activity in detergent and in liposomes and nanodiscs comprising a mixture of porcine brain polar lipids (BPL) and cholesterol (Chol) previously used to characterize several other human ABC transporters[32,33] (Fig. 1a and Supplementary Fig. 1a). Consistent with previously reported studies[34], ATP hydrolysis rates for ABCD1 were in the range of 10 nmol min$^{-1}$ mg$^{-1}$ in liposomes (not accounting for transporter orientation distribution) and nanodiscs, similar to values reported by other groups[34,35], and followed Michaelis Menten kinetics with Michaelis constant ($K_M$) values of about 0.3 mM ATP (Fig. 1a). Mutation of a catalytic glutamate residue to glutamine in the Walker B motif in the ABCD1 NBD (ABCD1$_{EQ}$) reduced ATPase rates by ~50%, similar to observations for ABCD4[26], and ATP hydrolysis was sensitive to inhibition by sodium orthovanadate (VO$_4$) or the non-hydrolysable ATP analog ATPγS (Supplementary Fig. 1b). We tested CoA esters of various VLCFAs at 0.1 and 1 mM concentrations for their ability to modulate the activity of ABCD1 as a readout for relevant substrate interactions in the absence of a direct transport assay. Detergent purified protein was employed to avoid any complications arising from VLCFA-CoA mediated membrane disruption in nanodiscs or liposomal samples. As shown in Fig. 1b, we observed ATPase rate stimulation for C24:0-CoA and C26:0-CoA, both reported to be substrates of ABCD1 and ABCD2[22], but not for C22:6-CoA or acetyl CoA alone when compared to control with no added CoA esters. Interestingly, we also observed ATPase rate stimulation by C18:1-CoA, suggesting that acyl chain saturation may play a key role in determining substrate specificity for ABCD1. To test whether the absence of stimulation in the presence of acetyl-CoA could be attributed to lack of a physical interaction with ABCD1, we tested the effect of acetyl-CoA and C24:0-CoA added together. Interestingly, this led to a marked decrease in the stimulatory effect of C24:0-CoA alone, suggesting competition from acetyl-CoA for the same binding site. The implications of these findings in the context of substrate transport are discussed below.

**Overall structure of nucleotide bound ABCD1**. To determine the structure of human ABCD1 in a lipid environment mimicking its physiological state, we reconstituted detergent purified ABCD1 in BPL/Chol nanodiscs utilizing membrane scaffold protein (MSP) 1D1 in the presence of the non-hydrolysable ATP analog ATPγS (Supplementary Fig. 1b). We obtained multiple structures from a single cryo-EM dataset (Supplementary Fig. 1d) but focused on two structures, a nucleotide bound OO state and an IO state resolved to 3.5 and 4.4 Å, respectively (Figs. 1c, d and 2a and Supplementary Figs 2, 3). While we observed a range of IO conformations with differing inter-NBD distances (Supplementary Fig. 1d), we focus here on the highest resolution of these. The quality of EM density for the higher resolution OO conformation allowed for *de novo* model building of a near complete model of ABCD1, residues 63−345, 380−433, and 462−684. Nucleotide bound ABCD1 adopts a characteristic exporter fold first identified in the bacterial ABC exporter Sav1866[36], entailing a domain-swapped architecture where transmembrane helix (TM) 4 and TM5, along with the intervening cytoplasmic helix (CH) from each protomer makes extensive contacts with the TMD and NBD of the opposite one (Fig. 1c). Each TMD contains six transmembrane helices that extend well beyond the membrane lipid bilayer in both directions. Two nucleotides, modeled as ATPγS, and two Mg$^{2+}$ ions are bound to the Walker A, Q-loop, and ABC signature motifs that exist within the interface between the two NBD (Fig. 1d). The OO conformation is characterized by a large cavity open to the peroxisome lumen and inner peroxisomal membrane. This cavity is lined by polar and charged

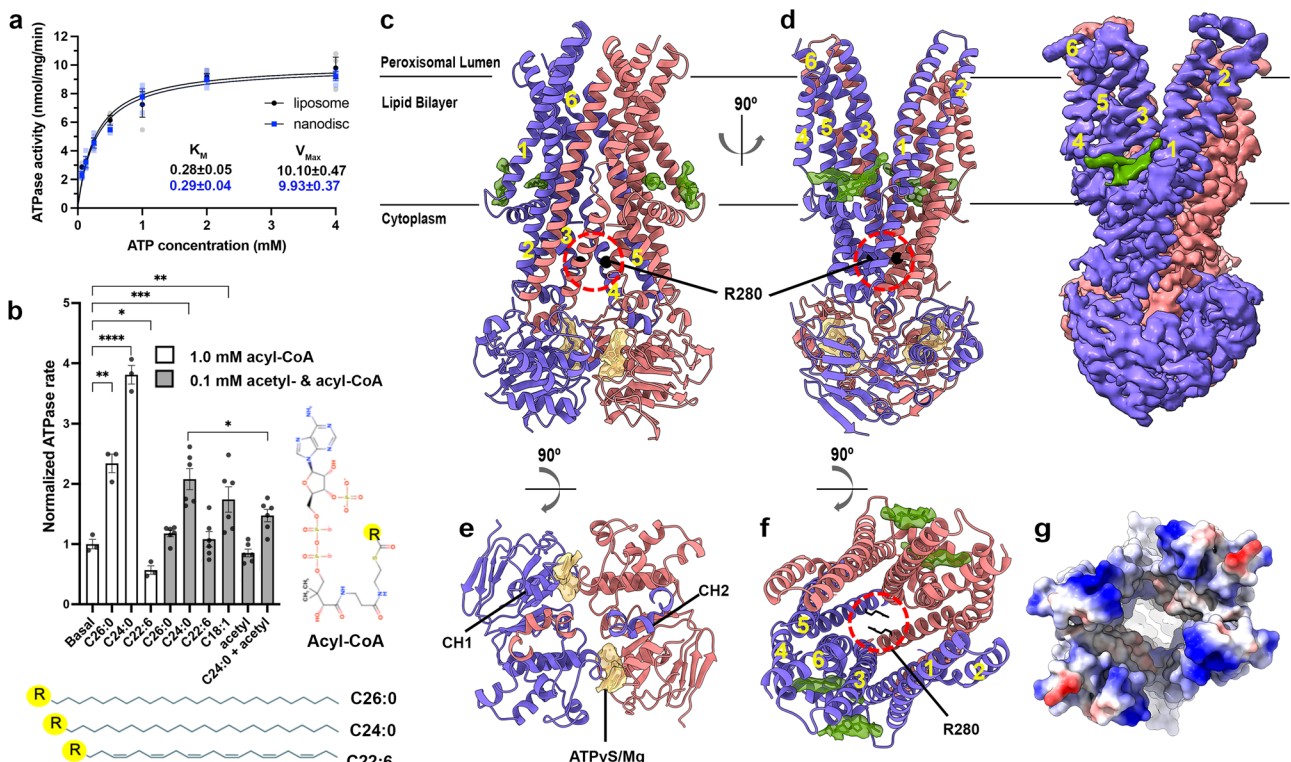

**Fig. 1 Structural and Functional characterization of outward open human ABCD1. a** ATPase activity of ABCD1 reconstituted in liposomes or nanodiscs as a function of ATP concentration, $n = 3$ and error bars represent standard deviation **b** ATPase activity of DDM/CHS detergent purified ABCD1 in the presence of CoA esters of the indicated varying acyl chain composition, normalized to ABCD1's basal rate at left, $n = 6$ (100 μM) or 3 (1 mM) and error bars represent S.E.M. Statistical significance by unpaired, two-tailed t-test $p$-values of <0.05, 0.01, 0.001, 0.0001 are indicated by *, **, ***, ****, respectively. **c**−**f** Map and structure of human ABCD1 outward open (OO) homodimer viewed from the membrane plane, made using only OO particles. EM density (0.04 contour) and ribbons are colored red and slate for each ABCD1 monomer, black sphere for R280, and green and yellow for modeled lipid-like entities and bound nucleotides, respectively. **e** Peroxisomal lumen with TMDs removed for clarity and **f** peroxisomal lumen with intracellular gate circled (dashed red line) with NBDs removed for clarity. TMs are numbered **g** electrostatic surface representation of ABCD1 viewed as a slice from the peroxisomal lumen.

residues from all 6 transmembrane helices from each protomer and is more hydrophilic and deeper than the corresponding cavities of ABCD4 or Sav1866 (Supplementary Fig. 4). It is also considerably wider, especially towards the opening to the peroxisomal lumen where TM5 and TM6 are splayed outward with a noticeable bend in TM6 and display considerable conformational disorder as judged by discontinuous EM density. At the other end of the cavity, a cytoplasmic gate formed by residue R280, stabilized by intrasubunit and intersubunit electrostatic interactions with surrounding residues, seals off access to the cytoplasm (Fig. 1f, g). The estimated cavity volume of ~28000 Å³ is large enough to accommodate phospholipids present in the surrounding nanodisc. However, no evidence of lipids was found inside the cavity, consistent with its overall hydrophilic nature. Interestingly, we observed lipid-like density features at the cytosolic membrane opening of the OO structure outside the TMD cavity that we tentatively modeled as a Chol and unspecified fatty acid acyl chains (Fig. 1c, d, f). One lipid density extends towards the gap between TM 1 and 3. The opening to the lumenal membrane in the OO cavity also revealed a similar, albeit weaker, shorter stretch of density consistent with another acyl chain (not modeled). As discussed below, we speculate that the acyl chains of VLCFA-CoAs could theoretically occupy similar positions while the CoA moieties bind within the TMD cavity during the substrate transport cycle.

**Transition from OO to IO conformations.** Although of lower resolution than that of the OO conformation, the EM map quality of the IO structure allowed for accurate placement of all TMs

(Fig. 2a). Density for the TMD region was of overall higher quality than that for the NBD (Supplementary Fig. 3), which were modeled by rigid body placement of the equivalent NBD from the OO structure without modeled nucleotides. The reduced resolution for the NBDs is likely due to increased conformational heterogeneity and is supported by the observation of several 3D classes leading to lower resolution IO structures with varying extents of NBD separation (Supplementary Fig. 1d). The final IO model comprised residues 67−443 and 460−684 and allowed for a direct comparison of the IO and OO structures from the same dataset (Fig. 2b). We observed greater continuity for TM5 and TM6 and the intervening extracellular region (Fig. 2a). This region's EM density was modeled as a short helical stretch in agreement with secondary structure prediction. Note that this helical stretch is comprised primarily of charged residues that are missing in ABCD2 and ABCD3.

Despite the large-scale overall conformational change, TM1-TM2, TM3-TM6, and TM4-TM5 pairs from the IO and OO conformations effectively move as rigid bodies, maintaining their overall conformation, except for a noticeable bend in TM6 in the OO conformation (Fig. 2c). These transitions follow conserved patterns previously described for type IV ABC transporters/type II exporters[37]. TM4 in both conformations contains a helical break around residues P263 and G266. G266 is conserved in all three peroxisomal ABCD transporters but not in ABCD4. The helical unwinding of TM4 was also found in ABCB1 and is related to the formation of an occluded conformation with bound substrates or inhibitors in association with its alternating-access mechanism[32]. A similar break in TM4 of the ABC transporter

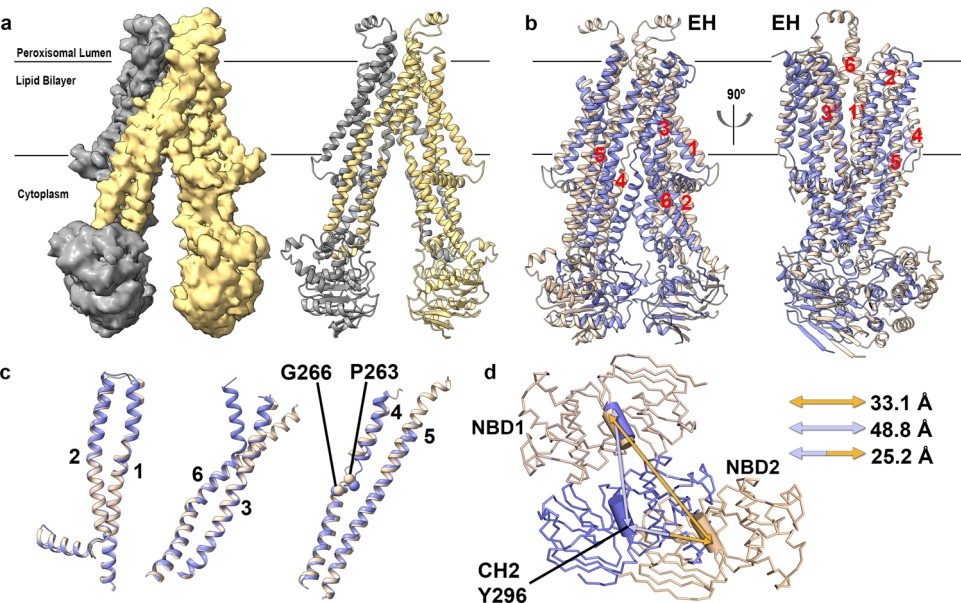

**Fig. 2 Comparison of IO and OO conformations of ABCD1. a** EM density map (0.04 contour) of human ABCD1 IO (cytoplasm facing) conformation colored yellow and gray for two monomers with similarly colored ribbon representing the protein backbone at right. **b** All Cα-atom overlay of IO and OO structures of human ABCD1 (gold and blue ribbons, respectively) with TMs numbered and external helix (EH) shown for IO conformation, with far viewing plane clipped to 10 Å. **c** Superposition of individual TM pairs from IO and OO structures with TM4 disrupting P263 & G266. **d** Comparison of NBDs of IO and OO structures aligned using NBD1 as reference from the lumenal viewpoint. Arrows depict inter coupling helix 2 (CH2) distances between Y296 Cα atoms.

YbtPQ was proposed as key for substrate release[38]. The OO-IO transition involves a 25.2 Å translation of the NBDs measured using inter CH2 distances with residue Y296 serving as a reference (Fig. 2d). The resulting conformation creates a large opening to the cytosol, disassembly of the cytoplasmic gating network via repositioning of a network of electrostatic and a few hydrogen-bond interactions that stabilize R280, and closure of the external cavity and accompanying formation of an external gate around D115, R389 and N390, that seals the cavity from the peroxisomal lumen (Fig. 3a, b). The larger estimated volume of the ABCD1 IO cavity is ~37000 Å$^3$ (Fig. 3c), both large enough to accommodate multiple acyl-CoA moieties. Like the cavity seen in the OO conformation, the IO cavity is also highly hydrophilic in nature and has openings to the outer peroxisomal membrane leaflet (Fig. 3d). As with the OO conformation, no evidence of bound lipids was found within the IO cavity either.

**Analysis of ABCD1 disease-causing mutations.** To date, over 800 mutations in the ABCD1 gene have been associated with X-ALD. Our ABCD1 structures allow us to pinpoint the location of several single amino acid substitutions and offer a basis for their associated pathogenic phenotype. Considering the high degree of sequential and structural conservation with ABC transporter NBDs, and the fact that substrate specificity largely derives from the TMD, here we focus solely on TMD mutations. These can be further divided into two subsets. The first, a cluster of mutations in TM1 and TM2, occurs at the opening to the inner peroxisomal membrane (Fig. 4a). It is plausible that this area is important for substrate transfer through the lateral inner peroxisomal membrane opening of ABCD1 in the OO conformation. A second cluster of mutations is comprised largely of residues from TM4 and TM5 and includes R280 and surrounding residues, which we speculate may lead to destabilization of the OO conformation and disruption of substrate transport. G266, also in TM4, is located at the lateral opening of ABCD1 to the outer peroxisomal membrane in the IO conformation (Fig. 4b). It is one

of the most frequently mutated residues in ALDP patients and is likely involved in allowing for secondary structure breaking and kinking of TM4. These data open the door for the design and execution of future in vitro mutagenesis studies aimed at analyzing the functional effects of select ABCD1 mutations.

## Discussion
In the absence of a substrate-bound state, the exact mechanisms of substrate recognition and translocation by ACBD1 remain elusive. However, our data offer insights into how VLCFA-CoA may be recognized, with the polar CoA moiety binding within a hydrophilic cavity open to the cytoplasm (IO conformation). The opening to the outer peroxisomal leaflet may offer a putative portal for the acyl chain of bound substrates to extend outside the cavity into the surrounding membrane space. The ATP-dependent switch to the OO conformation could entail a lateral movement of the acyl chain towards the inner peroxisomal membrane. The observation of lipid like density features parallel to the membrane plane (Figs. 1c, d, f and 4c) is in line with this speculative model, suggesting that acyl chains could, in principle, occupy similar positions during FA translocation. While the resolution of IO structure is insufficient to visualize lipids, superposition of the IO structure on that of the OO structure shows an inward opening cavity in relation to the observed density features (Fig. 4d). Moreover, our data show that while acetyl-CoA itself is unable to stimulate ATPase hydrolysis in ABCD1, it can partially inhibit ATPase stimulation by C24:0-CoA. Together, this points to both the binding of the CoA moiety to ABCD1 (to explain the competitive inhibition seen), and the requirement of the acyl chain (that acetyl-CoA lacks) to trigger the substrate-induced conformational changes promoting ATP binding and/or hydrolysis. Acyl chain flexibility may play a key role here, which is supported by our observation that unlike C26:0-CoA and C24:0-CoA, and C18:1-CoA, the less flexible C22:6-CoA, previously reported as an ABCD2 substrate[9,21,22], has no noticeable ATPase stimulation effect in ABCD1.

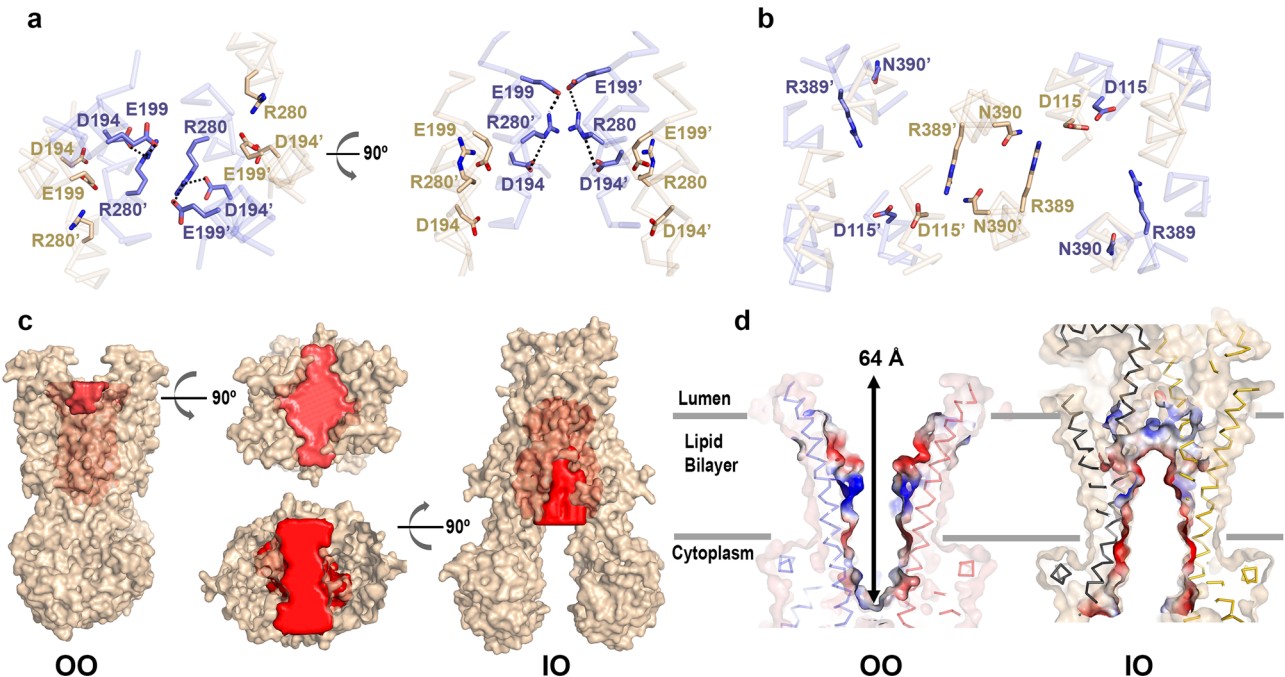

**Fig. 3 Comparison of IO and OO TMD cavities of ABCD1. a** All Cα-atom overlay of IO and OO ABCD1 structures (gold and blue sticks, respectively) but showing intracellular gate and surrounding residues viewed from the peroxisome lumen (left) and membrane plane (right). Equivalent residues from the two halves are distinguished by a prime symbol (**b**) same as (**a**) showing residues comprising the peroxisomal gate. **c** Surface representation of OO and IO structures of ABCD1 with solvent-exposed cavities colored red. **d** Slice through electrostatic surfaces of OO and IO structures of ABCD1 focused on the TMD region.

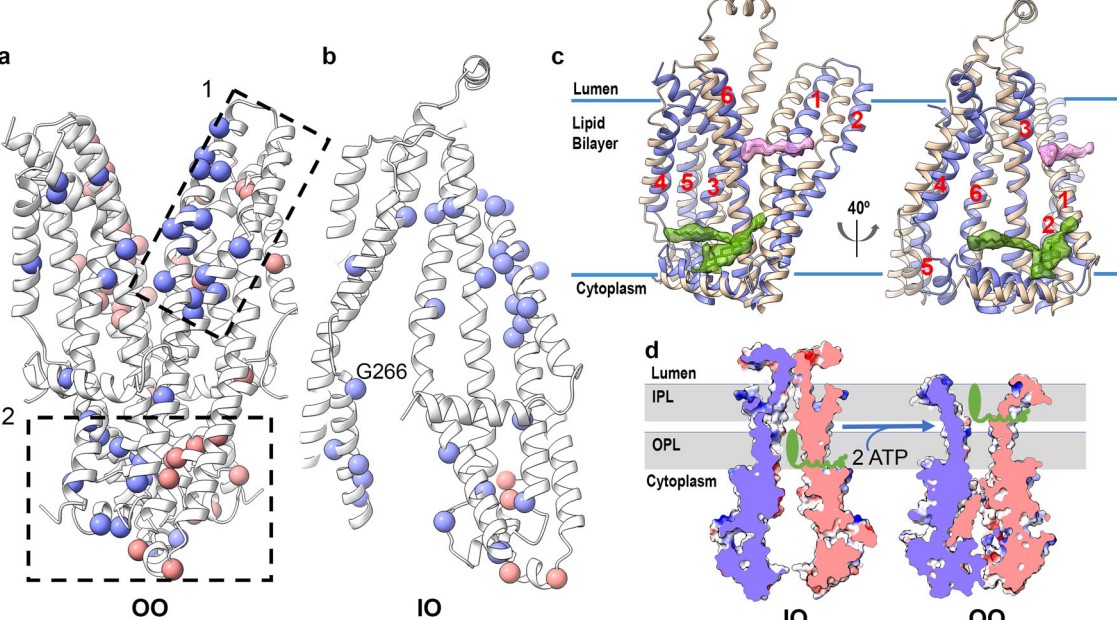

**Fig. 4 Analysis of TMD mutations in ABCD1 and proposed substrate interactions. a** ABCD1 TMD with Cα atoms of the most frequently mutated residues shown as spheres colored slate and pink for the two ABCD1 monomers and divided into two main clusters (dashed boxes 1 & 2). **b** Select ABCD1 disease-causing mutations mapped onto ABCD1 IO structure. **c** EM density (0.04 contour) of ABCD1 showing modeled lipid-like entities (green) observed near gap in the OO structure (blue), overlaid onto the IO structure (gold). Unmodeled lipid-like density is shown pink. **d** Hypothetical mechanism of ATP-dependent fatty-acyl-CoA (green) translocation across the peroxisomal membrane (gray) mediated by ABCD1.

It is currently unclear whether the fatty acyl chain is separated from its CoA ester for subsequent re-esterification, as has been proposed for the homologous transporter comatose/ABCD1 from *Arabidopsis thaliana*[35,39]. Our data lay the framework for future studies looking at the exact nature of a substrate-bound state of

ABCD1, characterization of its thioesterase activity, if any, and, if observed, delineating the mechanism whereby it may be tied to ATP hydrolysis dependent substrate transport. Our structures will also allow for the generation of more accurate models for ABCD2 and ABCD3 to shed light on the molecular mechanisms

that differentiate these three peroxisomal transporters, which share overlapping but distinct substrate transport profiles. Finally, our structures will prove valuable for the design of computational studies aimed at deciphering the fatty acid transport properties and lipid interactions of ABCD1 and provide a solid foundation for structure-based design of correctors or potentiators of ABCD1 for therapeutic use for X-ALD patients.

## Materials and methods

**Protein expression and purification**. We utilized the Flp-In TREX system (Thermo Fisher Scientific) to implement tetracycline-inducible expression of human ABCD1 to overcome several impediments associated with weak and inconsistent protein yields from transiently transfected HEK293T cells. Briefly, a synthetic gene construct of isoform 1 of human ABCD1 (Uniprot ID 095477) codon-optimized for human cell expression (GeneArt/Fisher Scientific) was first cloned in an expression vector comprising the pXLG gene expression cassette in a pUC57 vector backbone (GenScript) between BamHI and SalI restriction sites as described[40]. This added an eYFP-Rho1D4 purification tag preceded by a 3C/precision protease site at the C-terminal end of the construct. The full expression construct of ABCD1 or its EQ variant, generated through site directed mutagenesis using forward primer:

5′-CGGCCTAAGTACGCCCTGCTGGACCAGTGTACAAGCGCCGTGTCC ATCG-3′ and reverse primer: 5′CGTCGATGGACACGGCGCTTGTACACTGG TCCAGCAGGGCGTACTTAGG-3′ was transferred to a PCDNA5.1/FRT/TO vector between BamHI and NotI restriction sites and tetracycline-inducible stable cell lines were generated as per manufacturer's protocol. ABCD1 stable cells were grown in Dulbecco's Modified Eagle Medium (DMEM, Scientific) supplemented with 9% Fetal Bovine Serum (FBS, Gibco), penicillin/streptomycin mixture (Scientific), and antimycotic (Gibco) at 37 °C and 5% $CO_2$ under humidified conditions. For ABCD1 protein production, cells at a confluency of ~ 80% were induced with 0.6 μg/ml tetracycline in fresh DMEM supplemented with 2% FBS for an additional 24 h before being washed with Phosphate Buffered Saline (PBS), harvested, and flash frozen in liquid nitrogen. For $ABCD1_{EQ}$ expression, $ABCD1_{EQ}$ stable expressing cells were induced and harvested after 72-hour culture.

For ABCD1/$ABCD1_{EQ}$ purification, thawed cells were resuspended in a lysis buffer (25 mM Hepes pH 7.5, 150 mM sodium chloride (NaCl), 20% glycerol, and 1 mM Dithiothreitol (DTT, Scientific) supplemented with one Complete EDTA free protease inhibitor tablet (Roche) per 50 ml lysis buffer, 800 μM phenylmethylsulfonyl fluoride (PMSF, Sigma), and 20 μg/ml soybean trypsin inhibitor (Sigma). Cell lysate was dounce homogenized (8 strokes with the A pestle) and a 1% Dodecyl maltoside (DDM)/0.2% Cholesteryl hemisuccinate (CHS) (Anatrace) w:w mixture was added for protein solubilization. Protein extraction continued for 90 min at 4 °C with gentle agitation, followed by centrifugation at 48,000 r.c.f for 30 min at 4 °C. The supernatant was applied to a Rho-1D4 antibody (University of British Columbia) coupled to cyanogen bromide activated Sepharose 4B resin (Cytiva) and binding was allowed to proceed for 3 h at 4 °C. Beads were subsequently washed with a 4 × 10 bed volume (BVs) of wash buffer (25 mM Hepes pH 7.5, 150 mM NaCl, 20% glycerol, 0.02% DDM/0.004% w:w DDM/CHS, 1 mM DTT). Protein was eluted by incubation with 3 BVs of elution buffer (wash buffer supplemented with 3C protease using 1 milligram 3C per milliliter BV) overnight at 4 °C. The his-tagged 3C protease was removed by flowing and washing off cleaved transporter protein over two 100 μl beds of Ni-NTA Superflow resin in tandem (Qiagen). Eluent was concentrated using a 100 kDa cut-off Amicon Ultra filters (Millipore-Sigma). Gel lane 2 of Supplementary Fig. 1a (lane 9 of same uncropped gel in Supplementary Fig. 5) shows an example of detergent-solubilized ABCD1 purity.

**Nanodisc and liposome reconstitution**. Porcine BPL (Avanti) in chloroform was mixed with cholesterol (Chol; Sigma) at a final ratio of 80:20 w:w before being dried either under an Argon stream on ice or in a rotary evaporator (Büchi) before resuspension in Di-ethyl ether (Merck) and drying again. For nanodisc incorporation, detergent purified transporter lacking the fusion tag was mixed with membrane scaffold protein (MSP1D1, purified as described[41]), and an 80:20 w:w mixture of BPL and Chol containing 0.5% DDM/0.1% DDM/CHS using a 1:10:350 (ABCD1:MSP1D1:lipid mixture) molar ratio, and diluted with reconstitution buffer (25 mM Hepes pH 7.5, 150 mM NaCl) to reduce the glycerol concentration to 4% or lower. Nanodisc reconstitution proceeded for 30 min at room temperature (RT), followed by detergent removal using 0.8 mg pre-washed Biobeads SM-2 (Bio-Rad) per ml of reaction mix for 2 h at RT while slowly rolling. ABCD1 or ABC-$D1_{EQ}$ nanodiscs washed from Biobeads were concentrated using a 100 kDa Amicon Ultra centrifugal filter (Millipore-Sigma). SEC chromatogram in Supplementary Fig. 1a shows ABCD1 nanodisc quality.

For liposome preparation, detergent purified ABCD1 was mixed with a 80:20 w:w BPL/Chol lipid mixture at a protein:lipid ratio of 1:10 w:w, following established protocols[42] with minor changes. Briefly 0.14 and 0.3% Triton X100 (Sigma) was added to detergent purified ABCD1 concentrated to 1−1.5 mg/ml and BPL/Ch mix, respectively, incubated for 30 min at room temperature before being mixed, incubated again for 60 min with gentle agitation. Detergent was removed

during successive incubation steps using 20 mg fresh Biobeads SM2 per ml reaction mix each time. The incubation steps were carried out with gentle rolling for 30 min at RT, 60 min at 4 °C, overnight at 4 °C, and 2 × 60 min at 4 °C. The suspension was then centrifuged at 80 K rpm for 30 min in an ultracentrifuge. The supernatant was discarded, and the liposomal pellet was washed once with 1 ml of reconstitution buffer per 1 ml original reaction volume. The centrifugation step was repeated, the supernatant discarded, and proteoliposomes resuspended in reconstitution buffer to a concentration of 0.5−1 mg ml⁻¹.

**ATPase assays**. Proteins either in 0.02% DDM/0.004% w-w DDM/CHS detergent or reconstituted in 80:20 w-w BPL/Chol nanodiscs and liposomes were used in molybdate based calorimetric ATPase assay[43] measured at 850 nm on a SYNERGY Neo 2 Multi-Mode microplate reader (BioTek). The assay was carried out at 37 °C in a 40 μl volume in 96-well plates. The reaction mixture contained 25 mM Hepes pH 7.5, 150 mM NaCl, 10 mM Magnesium chloride ($MgCl_2$), and the protein concentration range was 0.1−0.2 mg ml⁻¹. The assays were started by the addition of 2 mM ATP, followed by a 40- or 60 min incubation at 37 °C, and stopped by the addition of 6% SDS. For ATP $K_M$ determination, an ATP concentration range of 0.0625 to 8 mM was used. Some experiments included ABCD1 ATPase inhibitors, either 5 mM ATPγS (TOCRIS) or 5 mM orthovanadate and 10 mM $MgCl_2$. A higher 5 mM ATP concentration was used in experiments where ABCD1 was incubated with acetyl-coenzyme A (CoA) or very-long-chain fatty acid-CoAs (VLCFA-CoAs: C26:0-CoA, C24:0-CoA, and C22:6-CoA). Two different VLCFA-CoA concentrations (i.e., 1 mM and 100 μM). Statistical analysis was done using GraphPad Prism 9. ATPase rates were determined using a simple linear regression to obtain a scalar value, and normalized to the basal ABCD1 ATPase rate. The $K_M$ and $V_{max}$ of nanodisc and liposome reconstituted ABCD1 were determined from the fit to the Michaelis-Menten equation of the corresponding ATPase rates. All assays were done at least in triplicate of three independent experimental setups. In assays with substrates, the means included three replicates and six replicates for experiments with 1 mM and 100 μM VLCFA-CoAs, respectively. Statistical significance was calculated by unpaired, two-tailed t-test, with $p$-values of <0.05, 0.01, 0.001, 0.0001 indicated by *, **, ***, ****, respectively. All VLCFA-CoAs were purchased from Avanti Polar Lipids and resuspended as per manufacturer instructions. Protein concentrations were measured using gel densitometry analyzed in Image Studio (LI-COR Biosciences) using detergent purified proteins of known concentrations determined by A280 measurements as standards. Data were normalized to ATPase rate for each set (11.6 and 13.6 nmol mg⁻¹ min⁻¹ for 0.1 and 1 mM datasets, respectively).

**Cryo-electron microscopy grid preparation and data collection**. Nanodisc reconstituted ABCD1 was further purified by size exclusion chromatography (SEC) on a G4000swxl SEC column (TOSOH biosciences) pre-equilibrated with reconstitution buffer on an Agilent 1260 Infinity II LC system (Agilent Technologies) at 4 °C. Peak fractions were pooled and mixed with 5 mM ATPγS (Sigma) and 10 mM $MgCl_2$ for 20 min at RT and concentrated to 0.5−1 mg ml⁻¹. A volume of 4 μl protein samples was applied to glow discharged (60 s, 15 mA current) Quantifoil R1.2/1.3 grids (Electron Microscopy Sciences, Hatfield, PA, USA) using a Vitrobot Mark IV (Thermo Fisher Scientific) with a 4 s blotting time and 0 blotting force under >90% humidity at 4 °C, then plunge frozen in liquid ethane.

Images were collected on a 300 kV Titan Krios electron microscope equipped with a Falcon 3EC direct electron detector (Thermo Fisher Scientific), with condenser C2 and 100 μm objective apertures. Automated data collection was carried out using the EPU 2.8.0.1256REL software package (Thermo Fisher Scientific) at a nominal magnification of 96,000×, corresponding to a calibrated pixel size of 0.895 Å with a defocus range from −0.8 to −2.6 μm. One shot was taken per hole. Image stacks comprising 60 frames were recorded for 60 s at an estimated dose rate of 1 electron/Å²/frame.

**Image processing**. All data processing steps were carried out within Relion 3.1 beta and Relion 3.1[44,45]. Motion correction was done using Relion's implementation of MotionCor2[46] and contrast transfer function (CTF) correction was done using Relion's gctf 1.06[47] wrapper. For initial processing, 3,402,440 particles picked from 8378 CTF corrected micrographs extracted at a 3× downscaled pixel size of 2.685 Å/pixel and subjected to multiple rounds of 2D classification. 538,915 particles selected after 2D classification were used for 3D classification (number of classes, $K = 5$) using a rescaled map of the cryo-EM structure of ABCD4[26] (EMDB-9791) as reference. C2 symmetry was applied for all 3D classification and refinement steps. 2 main classes were obtained with NBDs dissociated (IO) or dimerized (OO). The latter was refined to 7.5 Å and used as a reference for 3D classification for the full data set comprising 9040835 particles picked from 19695 image stacks from multiple data collections separated into subset 1 and subset 2. A flow chart depicting the subsequent data processing steps is shown in Supplementary Fig. 1d. For subset 1, 2845922 particles after 2D classification were used for 3D classification ($K = 8$) using our initial ABCD1 map as a 3D reference. The four highest resolution classes (highlighted blue) were reclassified ($K = 5$) to yield 2 main 3D classes for the OO (highlighted green) and IO (highlighted yellow) conformations. The former was refined to 8.3 Å resolution and the particles from this class were subjected to additional 2D classification from which 257981 selected particles were

used for further 3D classification ($K = 8$). The highest resolution of these was combined with an equivalent class from dataset 2 processed similarly to dataset 1. 64750 combined particles were refined to 5.5 Å resolution before being re-extraction using refined coordinates at a pixel size of 0.895 Å/pixel. After 3D refinement and Bayesian polishing, a 3.8 Å resolution map was obtained. Signal subtraction was then done to remove the nanodisc belt before an additional round of 3D classification ($K = 3$). 37237 particles from the highest resolution class was refined using original particles, followed by B-factor sharpening and post processing to yield a final map (Map 1) at 3.5 Å resolution. The refined map was used for mask generation for post-processing (soft edge of 8 pixels). A local resolution filtered map was generated using Relion's own algorithm.

The IO class from dataset 1 was refined to 7 Å resolution before being further 3D classified ($K = 8$). The three highest resolution classes from this round of 3D classification were combined with 2 IO classes from Dataset 2 processed similarly to Dataset 1 and further 3D classified ($K = 5$). 3 highest resolution classes were subjected to 3D refinement and Bayesian polishing to generate a 5.5 Å map. 2 similar classes from another round of 3D classification ($K = 3$) were further refined and the corresponding particles re-extracted to a new pixel size of 1.79 Å/pixel using refined coordinates and subjected to a final round of Bayesian polishing, 3D refinement, B-factor sharpening and post processing using a mask generated from the refined map (soft edge of 8 pixels) to yield a 4.4 Å map (Map 2) for the IO conformation.

**Model building**. Model building was done in Chimera 1.13.1[48] and coot 0.9.5[49,50]. Map 1 and its local resolution filtered version were used to build a model of the OO conformation of ABCD1 in coot using a SWISS-MODEL[51] generated homology model of ABCD1 based on the published ABCD4 structure as a starting point. The quality of EM density allowed accurate side chain placement for the bulk of the NBD and TMD residues (Supplementary Fig. 2). Both the post processed map and its local resolution filtered version were used for model building and real space refinement in Phenix 1.19.1[52]. For the IO conformation, TM1-TM2, TM4-TM5, and TM3-TM6 pairs and NBDs from the OO conformation structure were individually rigid body placed into Map 2 followed by manual adjustment as allowed for by the map, then refined in Phenix (Supplementary Fig. 3). The region between residues 361 and 370 was modeled as an alpha helix in agreement with secondary structure predictions. Figures were prepared in Chimera, ChimeraX 1.2.5[53], and The PyMOL 2.4.1 Molecular Graphics System (Schrödinger, LLC). Final data and coordinate statistics are in Supplementary Table 1 for both ABCD1 OO and IO structures. Secondary structure assignments are depicted on a sequence alignment of human ABCD1-4 in Supplementary Fig. 6.

**Statistics and reproducibility**. The sample sizes and statistical analyses used are presented in the legend of each figure.

**Reporting summary**. Further information on research design is available in the Nature Research Reporting Summary linked to this article.

## Data availability
The cryo-EM Maps for nanodisc reconstituted human ABCD1 have been deposited at the Electron Microscopy Databank (EMDB) under accession codes EMD-24656 (Map 1, ABCD1-OO) and EMD-24657 (Map 2. ABCD1-IO) respectively. Raw ABCD1 ATPase hydrolysis data is available in the file Supplementary Data 1. The associated atomic coordinates have been deposited at the Protein Data bank (PDB) under accession codes 7RR9 (ABCD1 OO) and 7RRA (ABCD1 IO).

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

## Acknowledgements

We would like to thank the cryo-EM and shared instruments core facilities at the Hormel Institute for help with experimental setup and Dr. Rhoderick Brown, Dr. Jarrod French, and Dr. Jeppe Olsen for critical reading and discussion during manuscript preparation. This work was supported in part by the Hormel Foundation (Institutional research funds to A.A.), the United Leukodystrophy Foundation (ULF research grant to A.A.), and the EAGLES Cancer Telethon Postdoctoral Fellowship (to L.T.M.L.).

## Author contributions

A.A. conceived the research. L.T.M.L., J.R.T., and A.A. performed all research with technical assistance from P.X.D. J.B. helped with EM sample preparation, handling, and data collection. L.T.M.L., J.R.T., and A.A. wrote the manuscript with input from all other authors.

## Competing interests

The authors declare no competing interests.
