## [Peer Review File · Communications Biology]

Reviewers' comments:

Reviewer #1 (Remarks to the Author):

In this report by Le and colleagues, the authors determine structures of the peroxisomal fatty acid transporter ABCD1 in inward-facing and outward-facing conformations. The structures reveal large cavities in both conformations that could accommodate long chain CoA lipid groups during transport. While the authors do not resolve CoA lipids in the structure, they demonstrate that acetyl-CoA can compete with CoA lipids. Based on this observation, the authors propose that there is a specific CoA binding site on the cytoplasmic side of the transporter. This work is clearly presented and suitable for publication in Communications Biology.

Reviewer #2 (Remarks to the Author):

Le et. al describes the purification, characterization and first cryo-EM structure determination of the human ABCD1 (very long chain fatty acid) transporter; a peroxisomal membrane protein whose dysfunction is responsible for causing X-linked adrenoleukodystrophy. The affinity tagged protein was expressed in HEK-293T cells, purified in detergent and reconstituted into MSP1D1 nanodiscs, where it displayed comparable activity to in liposomes, with a preference for C24:0-CoA as a substrate, and was taken forward for single particle cryo-EM analyses. Structures of the functional homodimer at 3.5 Å and 4.4 Å in the nucleotide-bound outward open, and apo inward open states indicated a large conformational change, and allowed comparison to related transporters, mapping of disease-causing mutation locations, and speculation about the molecular mechanism of transport.

This work has been methodically performed, and the manuscript was an enjoyable read and generally well-presented. The authors have importantly taken good care to ensure that the human ABCD1 transporter used for structural determination is highly pure, and displays good activity. Considering the novelty of structures for ABCD1, this work provides very important contribution to the field in terms advancing both understanding of the ABCD1 architecture, movement and disease mutation location, as well as informing about possible mechanism, and providing a basis for the design of targeted drugs. The methods section is detailed and would allow replication, with the minor additions mentioned below.

Specific comments:

1. In the methods section it isn't clear whether ATP hydrolysis was measured by endpoint assay (possibly implied by SDS quenching after 40-60 mins) or kinetically over that time course (as suggested by the use of linear regression) please clarify this, and comment in the text on the linearity of the reaction over the long measurement time (is it linear for that long with both high and low concentrations of ATP?).
2. In Fig 1A one label says D1-DDM, and I am not sure what that means, should this say no acyl chain control or something similar, or does this sample have DDM added when the others do not?
3. In Figure 1B, please apply statistical significance analyses between the different acyl-coA chains and control.
4. Fig 1C-E are ABCD homodimer, but is this the map from combined particles of OO plus IO, or just the OO class?
5. Lipid density is observed in what is described as the outer peroxisomal opening, however from the figures it appears that this density is only on the outside of the bundle, and does not look like it enters into the cavity formed by the dimer. Is the interpretation that the unresolved lipid tail might extend into the gap observed in fig 1C right between TMH3 and 1?

6. In the protein preparation, cholesterol hemisuccinate was added, but cholesterol (CLR) was modelled in a PDB instead, is there a reason for this?
7. Also in a PDB there is Lauryl dimethylamine-oxide (LDA) modelled, however this detergent does not appear to have been added to the purification procedure. I suggest asking the PDB annotator to change the identity of the ligand to something that is known to be present in the sample.
8. In the cryo-EM methods please add the following information:
 - Duration and current for glow discharge
 - Number of shots per hole (was aberration-free image shift used?)
 - Objective and C2 apertures used
 - EPU version
9. Please indicate how the mask for postprocessing was generated, and what soft edge was applied.
10. As this was a nanodisc preparation, was any orientation bias observed? If so please show this in a figure.
11. In lines 384-385 please alter the wording because it presently sounds like automated refinement was not performed after manual adjustment of the IO model.
12. Add in the software versions of GCTF, Coot and PHENIX used.
13. In Figure 1B the rates are normalized so please indicate in the methods or text what the control rate was.
14. I found the presentation style in Figure 2 B-D pretty unclear. It is not easy to see what has changed, so I suggest using stronger colours and less depth shading. For panel B, some curved arrows can be added to indicate the movement of the NBD domains. For C, I suggest a different representation, maybe cartoon or cylinder, and for D, there needs to be more of a description about what this is showing, and labelling the "intercoupling helices" in panel B would make it easier for reference. I think this is being shown from the cytoplasmic side, and that the top monomer is the one that is aligned, but how about labelling the monomers so it is clearer? As for my suggestion for fig 2B, a curved arrow showing which monomer is moving would aid quick interpretation.
15. Fig 3A – please specify which part has been aligned for the overlay.
16. For disease-causing mutations, only those in the TMD are discussed and shown. How many mutations are found in the NBD compared to the TMD, and are any of these in the nucleotide binding site or in residues important for dimerisation binding interactions?
17. In Supp Table 1, please add rama-z scores and EMRinger scores.
18. At the end of classification, ~1% and 0.4% of the starting particles were used for the final reconstructions. Some of the picked particles were certainly junk, but what were the reasons for taking forward only the selected classes in the second round of 3D classification, when from the figure, there seem to be other reasonably good-looking classes that were discarded. Were any of these investigated in case they contained additional protein conformations? In Supplementary Fig 1 there is a typo where one class contains 115.2% of particles that needs amending!

Reviewer #3 (Remarks to the Author):

In this manuscript, Alam and colleagues report cryo-EM structures of human ABCD1 transporter, which is responsible for import of very long chain fatty acids (VLCFAs) into peroxisomes. This

protein carries out an important function in the lipid metabolism, defects of which cause a human disease known as X-ALD, but the molecular mechanism by which ABCD1 and related members translocate VLCFAs across the peroxisomal membrane remains poorly understood. Both outward- and inward- open conformations of ABCD1 were captured in the authors' cryo-EM analysis, although the resolution of the inward-open structure is significantly lower (4.4 Å) than that of the outward-open structure. The inward-open structure probably did not resolve side chains of the protein, but amino acid registers are likely reasonably accurate as the conformational switch between the two structures seems to be mediated largely through rigid-body motions. The authors' working model suggesting a lipid flippase-like mechanism is interesting, but this model currently remains speculative because it is unclear how the large cavity observed in the TMD of ABCD1 is used for substrate transport. Of course, a substrate-bound structure and mutational analysis will be necessary to shed light on these issues, but such experiments could be technically challenging and thus probably more appropriate as the future investigations. Given that the presented structures are the first one for the VLCFA ABC transporters, the mechanistic insights provided here would be of interest to researchers in the field. Thus, I am supportive of publication under the condition that the authors address the points I listed below.

1. What is known about the fatty acid (FA) specificity of ABCD1? The term VLCFA suggests that the transporter preferentially translocates fatty acids with a very long chain. In Fig. 1 A and B, only very long chain substrates were used. Can the authors test shorter chain FAs to recapitulate the known specificity of ABCD1 with the purified protein?
2. The observation that C22:6 does not stimulate the ATPase activity of ABCD1 is interesting, but are there any previous report showing this type of FAs are not the substrates of ABCD1? This must be discussed in the main text.
3. In the introduction, the authors should describe more about the substrates of ABCD1, such as the specificity (related to points #1 and 2) and structural requirements. The current version is unclear about whether free FAs can be transported by ABCD1 or FAs must be CoA ester forms to be recognized by ABCD1. This information is necessary for nonexperts.
4. In the main text, the authors should clearly mention whether amino acids could or could not be unambiguously registered into the map of the inward-open structure, how the model was generated, and what are limitations in terms of interpretations. In this regard, it may be better to trim off all poorly resolved side chains from the final PDB models. Also please include panels of segmented TMs (overlay of the model and density map) in Supplementary Figure 2 to show the quality of the maps.
5. Clash scores are too high. The authors should improve the models to keep the score below 10.
6. Lines 118-122: I could not understand this statement clearly. Please split the sentence and describe what it means better in Figure 1.
7. Figure 1 C and E are too busy. The map/model overlay does not work well here. It would be clearer with the model only. The map can be presented as separate panels in Fig. 1 (if space is allowed) or a supplementary figure.
8. Figure 1E. The vertical position of R280 is not clear. Also indicate the position in panel C and F.
9. Line 152. Supplementary Figure 1D is too small to evaluate the flexibility of NBD. This should be presented as another supplementary figure.
10. Line 164. It is unclear where P263 and G266 are located. Indicate their locations in the relevant figure(s).
11. Lines 166-167. This statement is also difficult to understand. It probably needs more contextual information. Also, the sentence(s) should be clearer.
12. Lines 170-173. It is unclear what "electrostatic stabilization of R280" means.
13. Fig. 3. CoA is a relatively small structure, but Fig. 3C shows a seemingly very large cavity in both the OO and IO structures. I wonder whether the size of CoA and the cavities are compatible. The authors should mention about the comparison in the main text.
14. Fig. 4 and lines 205-207. I do not understand how the authors mapped the lipid-like density features from the OO structure onto the IO structure. What is the physical basis of this mapping? It seems to me that only property one can use is the vertical position along the membrane axis. Related to the model (Fig. 4D), the lipid-like densities shown in Fig. 1C seems irrelevant to the mechanism. Perhaps only the feature (one in the luminal leaflet) additionally shown in Fig. 4C is relevant to the model. If the authors agree with this, it is better to emphasize only this feature in the main figures (e.g., use of a different color). Also magnified panels should included because the current versions are too small to look at. Does one end of this lipid/detergent feature point into the

gate (towards the cavity)? If not, I would not be convinced whether this lipid/detergent feature is relevant at all.

15. Supplementary Fig. 3. ABCD1 and ABCD4 share significant sequence similarity, but they have different substrate specificity. The author should show a comparison between the structures of ABCD1 and ABCD4 and discuss where the different substrate specificity might originate.

16. Lines 214-215. This statement is confusing. It needs more contextual information (probably also related to points #1 and 3).

17. The current proposed model is speculative. In the discussion, the authors should clearly indicate the speculative nature of their model.

Minor points:

1. Line 71: "it's" \diamond "its".

2. Line 83: "ABCD1 in vitro activity" \diamond "In-vitro ATPase activity of ABCD1"

3. Line 97: "interactions of in the absence" \diamond "interactions in the absence"

4. Fig. 1B: "D1-DDM" should be replaced with "protein only" or defined in the legend.

5. Fig. 1B needs to be larger (especially acyl-CoA diagram is illegible).

6. Supplementary fig. 1 A and B are illegible too.

7. Supplementary fig. 1B: when experiments were not performed. It should be indicated with "N.D." It is unclear whether there is a bar with a near-zero value or no bar at all for "D1EQ". D1EQ needs to be defined in the legend.

Title

Structures of the human peroxisomal fatty acid transporter ABCD1 in a lipid environment

Authors

Le Thi My Le, James Robert Thompson, Phuoc Xuan Dang, Janarjan Bhandari & Amer Alam*

Reviewers' comments:

We thank the reviewers for their perceptive comments and suggestions for the improvement of the manuscript. We have adjusted our manuscript, followed by a point-by-point response to the reviewer comments.

Reviewer #1 (Remarks to the Author):

In this report by Le and colleagues, the authors determine structures of the peroxisomal fatty acid transporter ABCD1 in inward-facing and outward-facing conformations. The structures reveal large cavities in both conformations that could accommodate long chain CoA lipid groups during transport. While the authors do not resolve CoA lipids in the structure, they demonstrate that acetyl-CoA can compete with CoA lipids. Based on this observation, the authors propose that there is a specific CoA binding site on the cytoplasmic side of the transporter. This work is clearly presented and suitable for publication in Communications Biology.

We appreciate the reviewer's positive critique of our paper.

Reviewer #2 (Remarks to the Author):

Le et. al describes the purification, characterization and first cryo-EM structure determination of the human ABCD1 (very long chain fatty acid) transporter; a peroxisomal membrane protein whose dysfunction is responsible for causing X-linked adrenoleukodystrophy. The affinity tagged protein was expressed in HEK-293T cells, purified in detergent and reconstituted into MSP1D1 nanodiscs, where it displayed comparable activity to in liposomes, with a preference for C24:0-CoA as a substrate, and was taken forward for single particle cryo-EM analyses. Structures of the functional homodimer at 3.5 Å and 4.4 Å in the nucleotide-bound outward open, and apo inward open states indicated a large conformational change, and allowed comparison to related transporters, mapping of disease-causing mutation locations, and speculation about the molecular mechanism of transport.

This work has been methodically performed, and the manuscript was an enjoyable read

and generally well-presented. The authors have importantly taken good care to ensure that the human ABCD1 transporter used for structural determination is highly pure, and displays good activity. Considering the novelty of structures for ABCD1, this work provides very important contribution to the field in terms advancing both understanding of the ABCD1 architecture, movement and disease mutation location, as well as informing about possible mechanism, and providing a basis for the design of targeted drugs. The methods section is detailed and would allow replication, with the minor additions mentioned below.

Specific comments:

1. In the methods section it isn't clear whether ATP hydrolysis was measured by endpoint assay (possibly implied by SDS quenching after 40-60 mins) or kinetically over that time course (as suggested by the use of linear regression) please clarify this, and comment in the text on the linearity of the reaction over the long measurement time (is it linear for that long with both high and low concentrations of ATP?).

Answer: Lines 321-323: We used end-point assays. As stated in the manuscript already, ATPase hydrolysis was stopped by addition of 6% SDS after either a 40- or 60-minute incubation period.

Lines 329-331: We clarified in Methods that we used a simple linear regression to obtain a scalar constant for basal rate normalization shown in Fig. 1B.

For our initial tests, we established linearity using three time points of 0, 30 and 60 min for ABCD1 reconstituted into BPL/Ch nanodiscs using 2 mM ATP (see plot below) . The rate obtained ($7.6 \text{ nmol min}^{-1} \text{ mg}^{-1}$ with std error of 0.7), was in reasonable agreement with our subsequent ATP hydrolysis rates reported in the manuscript (Fig. 1A). As this was a test that established our reaction conditions, the plot below is not included in the manuscript.

2. In Fig 1A one label says D1-DDM, and I am not sure what that means, should this say no acyl chain control or something similar, or does this sample have DDM added when the others do not?

Lines 329-331 & 584-595: We regret that our presentation of D1-DDM was unclear and now address this in our revision. D1-DDM has been changed to "Basal", meaning ABCD1

control purified in detergent 0.02%DDM/0.004%CHS without any acyl-CoA added. See Fig. 1B below.

3. In Figure 1B, please apply statistical significance analyses between the different acyl-coA chains and control.

Lines 336-337 & 584-595: In our revised manuscript we present the requested statistical significance analyses. We added acyl C18:1-CoA data to this figure (related to reviewer 3's point #1).

Fig. 1B

4. Fig 1C-E are ABCD homodimer, but is this the map from combined particles of OO plus IO, or just the OO class?

Figure 1 only pertains to the OO class.

Lines 591: Added to legend for Fig. 1D that map shown is made from the OO class only.

5. Lipid density is observed in what is described as the outer peroxisomal opening, however from the figures it appears that this density is only on the outside of the bundle, and does not look like it enters into the cavity formed by the dimer. Is the interpretation that the unresolved lipid tail might extend into the gap observed in fig 1C right between TMH3 and 1?

Lines 212-221 & 584-595 & 653-655: These lipid densities do not enter the cavity but are located at a gap leading into the cavity. We have clarified this in our manuscript and added Fig. 4C, an overlay of the ABCD1 OO (with its lipid like densities shown) aligned to the IO conformation structure. Our assertion is that the observation of these densities is compatible with the possibility that the acyl chains of substrates may extend outside the actual TMD cavity, though we do not have a structure for the protein:substrate complex. We speculate that a portion of the acyl-chain might reside outside the cavity during VLCFA-CoA binding based on these observations of density features parallel to the membrane plane. .

6. In the protein preparation, cholesterol hemisuccinate was added, but cholesterol (CLR) was modelled in a PDB instead, is there a reason for this?

The structure was determined in a lipid environment (i.e. nanodiscs), that comprises 80:20 w: w brain polar lipid: cholesterol. As such, we have modeled cholesterol instead of CHS.

7. Also in a PDB there is Lauryl dimethylamine-oxide (LDA) modelled, however this detergent does not appear to have been added to the purification procedure. I suggest asking the PDB annotator to change the identity of the ligand to something that is known to be present in the sample.

We apologize for this oversight and have fixed the error. LDA was replaced by a C12 acyl chain and marked UNL in the PDB file.

8. In the cryo-EM methods please add the following information:
- Duration and current for glow discharge

Line 350: We added these details to paper: glow discharge (60 sec, 15 mA current).

- Number of shots per hole (was aberration-free image shift used?)

Line 359: The requested information has been added to the methods section. A single shot/hole was used.

Aberration-free image shift was not used.

- Objective and C2 apertures used

Lines 355-357: We added to Methods these details: EPU version 2.8.0.1256REL and a condenser C2 (50) aperture and an objective aperture were used.

9. Please indicate how the mask for postprocessing was generated, and what soft edge was applied.

Line 388 & 397-398: We added to Methods these details: The mask for post processing was generated in Relion with a soft edge of 8 pixels.

10. As this was a nanodisc preparation, was any orientation bias observed? If so please show this in a figure.

The orientation distribution plots have now been added as Supplementary Figures 2 and 3 (panel C for both). We observed a proportionally higher number of side views for both conformations (see below).

Supplementary Figure 2C

Supplementary Figure 3C

11. In lines 384-385 please alter the wording because it presently sounds like automated refinement was not performed after manual adjustment of the IO model.

Line 410: We now report use of Phenix to refine the IO model following manual building.

12. Add in the software versions of GCTF, Coot and PHENIX used.

Lines 364, 401, 407: We added to Methods the details regarding the program versions used: GCTF 1.06, Chimera 1.13.1, coot 0.9.5, Phenix 1.19.1.

13. In Figure 1B the rates are normalized so please indicate in the methods or text what the control rate was.

Lines 341-343: The basal rate of D1-DDM ranged from 11.68 (100 μ M dataset) to 13.68 nMol $\text{mg}^{-1} \text{min}^{-1}$ (1mM dataset), which is similar, albeit slightly higher than the rates measured for ABCD1 incorporated in liposomes and nanodiscs.

14. I found the presentation style in Figure 2 B-D pretty unclear. It is not easy to see what has changed, so I suggest using stronger colours and less depth shading. For panel B, some curved arrows can be added to indicate the movement of the NBD domains. For C, I suggest a different representation, maybe cartoon or cylinder, and for D, there needs to be more of a description about what this is showing, and labelling the “intercoupling helices” in panel B would make it easier for reference. I think this is being shown from the cytoplasmic side, and that the top monomer is the one that is aligned, but how about labelling the monomers so it is clearer? As for my suggestion for fig 2B, a curved arrow

showing which monomer is moving would aid quick interpretation.

Lines 177-179 & 599-606: We regret that our presentation of Figure 2 was unclear and have updated the figure as per reviewer recommendations where appropriate. Specifically, we separated the cartoon model and map for Figure 2A, similar in format to the OO map image in Figure 1D. We made the superimposed ABCD1 OO and IO cartoon models non-transparent and now show two 90° rotated views. We now report that this structural alignment is over all-atoms within the figure legend. We used the suggested cartoon depiction for Fig. 2C. We added the CH2 label for the coupling helices used in 2D as the reviewer suggested.

Fig. 2.

15. Fig 3A – please specify which part has been aligned for the overlay.

Lines 625-628: Figure 3A and B shows the same D1 OO to IO superposition shown in Fig. 2B, but provides a magnified view focused on the intracellular and peroxisomal gate regions. We now report this alignment is over all-C α atoms within the legend for Fig. 3A-B.

16. For disease-causing mutations, only those in the TMD are discussed and shown. How many mutations are found in the NBD compared to the TMD, and are any of these in the nucleotide binding site or in residues important for dimerisation binding interactions?

The NBD domains and their dimerized sandwich is very well conserved in ABC transporters. As such, we chose to focus only on the TMD, which is responsible for ligand binding. This is clarified in the manuscript (Lines 192-195).

17. In Supp Table 1, please add rama-z scores and EMRinger scores.

The Rama-Z information has been added to Supplementary Table 1 for both structures. For the ABCD1 OO structure to 3.5Å, a 1.6 EMRinger value is now reported. For the low 4.4Å resolution IO structure with less clear sidechain densities, the EMRinger value is not determined.

Maps	Human ABCD1 OO	Human ABCD1 IO
MolProbity Statistics		
MolProbity Score	1.63	1.65
Clashscore	6.67	8.95
Poor rotamers (%)	0.00	0.00
Z-Score		
Whole	1.00	1.98
Helix	1.41	2.17
Sheet	1.08	1.98
Loop	-1.53	-1.51

Relevant portion of Supplementary Table 1

18. At the end of classification, ~1% and 0.4% of the starting particles were used for the final reconstructions. Some of the picked particles were certainly junk, but what were the reasons for taking forward only the selected classes in the second round of 3D classification, when from the figure, there seem to be other reasonably good-looking classes that were discarded. Were any of these investigated in case they contained additional protein conformations? In Supplementary Fig 1 there is a typo where one class contains 115.2% of particles that needs amending!

The good-looking classes that the reviewer mentions mostly belonged to different IO conformations. Our attempts to obtain a spectrum of additional near atomic resolution IO structures, however, did not go beyond a number of lower resolution (8-10 Å) structures, an analysis of which would not contribute to any further mechanistic insights. As such, we chose to focus on the highest resolution of these.

We thank the reviewer for highlighting the error in Supplementary Fig 1, which has been corrected to read 15.2% (Supplementary Figure 1D).

Reviewer #3 (Remarks to the Author):

In this manuscript, Alam and colleagues report cryo-EM structures of human ABCD1 transporter, which is responsible for import of very long chain fatty acids (VLCFAs) into peroxisomes. This protein carries out an important function in the lipid metabolism, defects of which cause a human disease known as X-ALD, but the molecular mechanism by which ABCD1 and related members translocate VLCFAs across the peroxisomal

membrane remains poorly understood. Both outward- and inward- open conformations of ABCD1 were captured in the authors' cryo-EM analysis, although the resolution of the inward-open structure is significantly lower (4.4 Å) than that of the outward-open structure. The inward-open structure probably did not resolve side chains of the protein, but amino acid registers are likely reasonably accurate as the conformational switch between the two structures seems to be mediated largely through rigid-body motions. The authors' working model suggesting a lipid flippase-like mechanism is interesting, but this model currently remains speculative because it is unclear how the large cavity observed in the TMD of ABCD1 is used for substrate transport. Of course, a substrate-bound structure and mutational analysis will be necessary to shed light on these issues, but such experiments could be technically challenging and thus probably more appropriate as the future investigations. Given that the presented structures are the first one for the VLCFA ABC transporters, the mechanistic insights provided here would be of interest to researchers in the field. Thus, I am supportive of publication under the condition that the authors address the points I listed below.

1. What is known about the fatty acid (FA) specificity of ABCD1? The term VLCFA suggests that the transporter preferentially translocates fatty acids with a very long chain. In Fig. 1 A and B, only very long chain substrates were used. Can the authors test shorter chain FAs to recapitulate the known specificity of ABCD1 with the purified protein?

Lines 55-58: The issue of differential substrate specificities of peroxisomal ABC transporters has been reviewed previously (Morita and Imanaka 2012) and largely derive from complementation assays in yeast knockouts of the ABCD1 homologs pxa1/pxa2 (van Roermund, Visser et al. 2008, van Roermund, Visser et al. 2011). These studies established broad specificity for saturated fatty acyl-CoAs for ABCD1, while polyunsaturated fatty acyl Co-As like C22:6-CoA are thought to be ABCD2 substrates. While ABCD1 likely also transports medium chain fatty acids, it is, along with ABCD2, unique in its ability to transport specific VLCFAs (C22:0 or higher). Here we focused on the fatty acyl-CoAs whose cytosolic accumulation is documented in X-ALD patients.

Lines 106-108 & 226-228: As requested, we have now added data for the shorter C18:1-CoA to Fig. 1B, which also stimulates ABCD1 ATPase activity. Considering that C22:6-CoA does not stimulate activity, these data suggest that acyl chain saturation is a key distinguishing factor of ligand specificity, at least within the confines of our assay.

Fig. 1B

2. The observation that C22:6 does not stimulate the ATPase activity of ABCD1 is interesting, but are there any previous report showing this type of FAs are not the substrates of ABCD1? This must be discussed in the main text.

Lines 55-58 & 105: This is related to the reviewer's 1st point above. Acyl C22:6-CoA was reported to not be a ABCD1 substrate previously by others, but rather as a substrate of ABCD2. This point has been clarified in the Introduction.

3. In the introduction, the authors should describe more about the substrates of ABCD1, such as the specificity (related to points #1 and 2) and structural requirements. The current version is unclear about whether free FAs can be transported by ABCD1 or FAs must be CoA ester forms to be recognized by ABCD1. This information is necessary for nonexperts.

Lines 55-58: As stated above, we have added references for published data in our updated Introduction. Data using purified transporters is lacking and, in the absence of substrate bound ABCD1 structures, we believe inferring the structural requirements of ligand specificity would be too speculative.

4. In the main text, the authors should clearly mention whether amino acids could or could not be unambiguously registered into the map of the inward-open structure, how the model was generated, and what are limitations in terms of interpretation. In this regard, it may be better to trim off all poorly resolved side chains from the final PDB models. Also please include panels of segmented TMs (overlay of the model and density map) in Supplementary Figure 2 to show the quality of the maps.

We added the overlay of model and map for TMs to two new Supplementary Figures 2 & 3 as suggested, and added the map density individually for the NBD and ligands as well.

Supplementary Figure 2A - ABCD1 OO model and 3.5Å resolution map

Supplementary Figure 3A - ABCD1 IO model and its 4.4Å resolution map

However, we have chosen not to truncate any sidechain atoms from the coordinates, in consideration of the fact that the model was generated using the rigid body placements of relevant regions that follow conserved patterns for type IV ABC transporters (also known as Type 1 ABC exporters). Rather we remove the atoms for complete residues where the mainchain cannot be placed. This practice is increasingly common over the last few years with real-space refinement within phenix explicitly taking the reported resolution into account. Moreover, some programs cannot parse PDB files containing residues with incomplete sidechains.

5. Clash scores are too high. The authors should improve the models to keep the score below 10.

We have now improved the geometry of both models as suggested. The updated refinement statistics are reported in Supplementary Table 1.

Dataset	Nanodisc reconstituted ABCD1	
Magnification	96k	
Pixel Size (Å)	0.895	
Total Dose (e/Å ²)	60	
Defocus Range (um)	-0.8 to 2.6	
Maps	Human ABCD1 OO	Human ABCD1 IO
EMDB ID	EMD-24656	EMD-24657
# Particles in final Class	37237	91325
Resolution (Å) (0.143 threshold)	3.5	4.4
Sharpening B factor	-65	-50
Refined Coordinates	Human ABCD1 OO	Human ABCD1 IO
PDB ID	7RR9	7RRA
# Residues/Non-hydrogen Atoms	1120/9096	1204/9610
Ligands	8	
R.M.S deviations		
Bond Length (Å)	0.003	0.003
Bond Angles (°)	0.591	0.608
MolProbity Statistics		
MolProbity Score	1.63	1.65
Clashscore	6.67	8.95
Poor rotamers (%)	0.00	0.00
Z-Score		
Whole	1.00	1.98
Helix	1.41	2.17
Sheet	1.08	1.98
Loop	-1.53	-1.51
EM Ringer Score	1.6	N.D
Ramachandran statistics		
Favored (%)	96.12	96.99
Allowed (%)	3.88	3.01
Outliers (%)	0.00	0.00

Supplementary Table 1: Data processing and refinement statistics

6. Lines 118-122: I could not understand this statement clearly. Please split the sentence and describe what it means better in Figure 1.

Lines 118-121: We clarified this point, explaining there were multiple structures seen but focused on only two that refined to higher resolutions.

7. Figure 1 C and E are too busy. The map/model overlay does not work well here. It would be clearer with the model only. The map can be presented as separate panels in Fig. 1 (if space is allowed) or a supplementary figure.

We separated the model and map in Fig. 1C-D as shown here.

8. Figure 1E. The vertical position of R280 is not clear. Also indicate the position in panel C and F.

We clarified the position of R280 at the cavity bottom in Fig. 1F and Fig. 1C-D shown above. We rearranged the other Figure 1 panels in response to other reviewer comments.

Fig. 1F.

9. Line 152. Supplementary Figure 1D is too small to evaluate the flexibility of NBD. This should be presented as another supplementary figure.

We rearranged Supplementary Figure 1 to increase the size of Fig S1D.

10. Line 164. It is unclear where P263 and G266 are located. Indicate their locations in the relevant figure(s).

We now indicate this region and these two TM4 helix breaking residues in Fig. 2C.

Fig. 2C.

11. Lines 166-167. This statement is also difficult to understand. It probably needs more contextual information. Also, the sentence(s) should be clearer.

Lines 176-177: We clarified this distance error in the manuscript to correspond to the 25.2Å distance shown in Figure 2D.

12. Lines 170-173. It is unclear what “electrostatic stabilization of R280” means.

Lines 178-183: A statement has been clarified to indicate that there is a network of electrostatic interactions involving R280 and other residues as shown in Fig. 3A-B.

13. Fig. 3. CoA is a relatively small structure, but Fig. 3C shows a seemingly very large cavity in both the OO and IO structures. I wonder whether the size of CoA and the cavities are compatible. The authors should mention about the comparison in the main text.

Line 183-185: We agree with the reviewer that the cavities are quite large. This is not uncommon for Type IV ABC transporters such as ABCB1, which transports substrates of a wide range of shapes and sizes. From our structures of ABCD1, the cavity should be able to accommodate more than one CoA moiety. Based on the hydrophilic nature of the TMD cavities, we propose that the acyl chains extend out into the bulk membrane. As such, we see no reason that CoA would be incompatible with the observed TMD cavities in the IO or OO states. We also now report the calculated cavity volumes for both ABCD1 conformations.

14. Fig. 4 and lines 205-207. I do not understand how the authors mapped the lipid-like density features from the OO structure onto the IO structure. What is the physical basis of this mapping? It seems to me that only property one can use is the vertical position along the membrane axis. Related to the model (Fig. 4D), the lipid-like densities shown in Fig. 1C seems irrelevant to the mechanism. Perhaps only the feature (one in the luminal leaflet) additionally shown in Fig. 4C is relevant to the model. If the authors agree with this, it is better to emphasize only this feature in the main figures (e.g., use of a different color). Also magnified panels should included because the current versions are too small to look at. Does one end of this lipid/detergent feature point into the gate (towards the cavity)? If not, I would not be convinced whether this lipid/detergent feature is relevant at all.

Lines 212-221: We apologize for this point being unclear. We simply used the same all atom alignment to overlay the IO structure onto the OO structure, which is now shown in Fig. 4C. Based on this, the observed density features modeled as acyl chains would sit at the opening to the outer peroxisomal leaflet in the IO structure. In other words, it is compatible with our speculation that the CoA moiety would sit inside the TMD, while the acyl chain could extend out. This point has been clarified in the revised manuscript.

15. Supplementary Fig. 3. ABCD1 and ABCD4 share significant sequence similarity, but they have different substrate specificity. The author should show a comparison between the structures of ABCD1 and ABCD4 and discuss where the different substrate specificity might originate.

As requested, the equivalently aligned, OO conformations of ABCD1 and ABCD4, as well as of Sav1866, are included as Supplementary Figure 4. Our remade figure better shows a comparison of the electrostatic differences within each of these TMD cavities that bind substrates. However, in the absence of substrate bound structures, any comparison of other specific structural features that may underlie substrate specificity differences would be mere speculation. We do, however, point out the OO cavity in ABCD1 is significantly deeper than the analogous cavity in ABCD4 (Lines 133-136) and report the cavity volumes of both ABCD1 structures (Lines 183-185). Most importantly, as reported in original submission as well as in the revision, ABCD1 transports substrates in the opposite direction to ABCD4. As such, the OO conformation in ABCD1 is for substrate release to the peroxisomal lumen, while it is the IO conformation of ABCD4 that theoretically releases Vitamin B12 to the cytoplasm.

Supplementary Figure 4

16. Lines 214-215. This statement is confusing. It needs more contextual information (probably also related to points #1 and 3).

Lines 223-229: We attempted to clarify this interpretation of our ATPase activity comparison between acetyl-CoA data and either the C24-CoA or C26-CoA data, which indicates that ABCD1 clearly imposes a “requirement” for an acyl chain longer than an acetyl moiety to affect ATPase stimulation.

17. The current proposed model is speculative. In the discussion, the authors should clearly indicate the speculative nature of their model.

Lines 212-221: We have clarified further in the Discussion that our model is completely hypothetical, even using the phrase “speculative model” in line 217.

Minor points:

1. Line 71: “it’s” \diamond “its”.
2. Line 83: “ABCD1 in vitro activity” \diamond “In-vitro ATPase activity of ABCD1”
3. Line 97: “interactions of in the absence” \diamond “interactions in the absence”
4. Fig. 1B: “D1-DDM” should be replaced with “protein only” or defined in the legend.
5. Fig. 1B needs to be larger (especially acyl-CoA diagram is illegible).
6. Supplementary fig. 1 A and B are illegible too.
7. Supplementary fig. 1B: when experiments were not performed. It should be indicated with “N.D.” It is unclear whether there is a bar with a near-zero value or no bar at all for “D1EQ”. D1EQ needs to be defined in the legend.

Thank you. All these edits are corrected.

REVIEWERS' COMMENTS:

Reviewer #2 (Remarks to the Author):

The updated version of this manuscript now has clearer figures and better descriptive text, and the authors have addressed many of my concerns.

There are still some outstanding issues:

1. The EMRinger score will describe how well justified the model side chain detail is by the experimental density. As side chains are modeled for both IO and OO, poor side chain visibility is not a reason to exclude this measure. Kindly present EMRinger scores for both models as originally requested to allow evaluation of the reliability of the models.
2. Fig 2D (right). Please specify in the legend if this view is from the lumen or cytoplasm.
3. Fig 2B (right) has a floating red apostrophe - please attach it to its relevant number.

Reviewer #3 (Remarks to the Author):

Overall, the manuscript has been improved, and the authors' rebuttal is reasonable. Thus, I recommend the publication of this manuscript. I only have minor suggestions.

The terms "outer peroxisomal membrane" and "inner membrane" are inaccurate and can be confusing (examples: lines 145 and 148). The outer and inner membranes are typically to refer to those membranes of double-membrane organelles like the nucleus, mitochondria and chloroplasts. The accurate terms are outer (cytosolic) and inner (luminal) "leaflets of the membrane".

Line 220: "inward opening cavity" sounds strange.

Although these errors are minor, I suggest the authors read through the manuscript carefully to correct any errors. For example, "nMol mg-1 min-1" should be "nmol mg-1 min-1", "100" should be "100 μm " in line 355, "Post" should be "post" in line 397, and "Pixels" should be "pixels" in line 398.

Lines 659 and 660. Both "modeled lipid-like entities (green and pink)" and "Unmodeled lipid-like density is shown pink" cannot be right. In Figure 4C, both green and pink densities appeared to be modeled.

Title

Structures of the human peroxisomal fatty acid transporter ABCD1 in a lipid environment

Authors

Le Thi My Le, James Robert Thompson, Phuoc Xuan Dang, Janarjan Bhandari & Amer Alam*

Reviewers' comments:

We thank the reviewers for their perceptive comments and suggestions for the improvement of the manuscript. We have adjusted our manuscript, followed by a point-by-point response to the reviewer comments.

Reviewer #2 (Remarks to the Author):

The updated version of this manuscript now has clearer figures and better descriptive text, and the authors have addressed many of my concerns.

There are still some outstanding issues:

1. The EMRinger score will describe how well justified the model side chain detail is by the experimental density. As side chains are modeled for both IO and OO, poor side chain visibility is not a reason to exclude this measure. Kindly present EMRinger scores for both models as originally requested to allow evaluation of the reliability of the models.

The EM-Ringer value is now reported for the ABCD1 IO structure.

2. Fig 2D (right). Please specify in the legend if this view is from the lumen or cytoplasm.

This view is from the luminal viewpoint as indicated in the Fig 2D legend.

3. Fig 2B (right) has a floating red apostrophe - please attach it to its relevant number.

Thank you. It is now corrected.

Reviewer #3 (Remarks to the Author):

Overall, the manuscript has been improved, and the authors' rebuttal is reasonable. Thus, I recommend the publication of this manuscript. I only have minor suggestions.

The terms "outer peroxisomal membrane" and "inner membrane" are inaccurate and can be confusing (examples: lines 145 and 148). The outer and inner membranes are typically to refer to those membranes of double-membrane organelles like the nucleus, mitochondria and chloroplasts. The accurate terms are outer (cytosolic) and inner (luminal) "leaflets of the membrane".

Line 220: "inward opening cavity" sounds strange.

Although these errors are minor, I suggest the authors read through the manuscript carefully to correct any errors. For example, "nMol mg-1 min-1" should be "nmol mg-1 min-1", "100" should be "100 μ m" in line 355, "Post" should be "post" in line 397, and "Pixels" should be "pixels" in line 398.

Lines 659 and 660. Both "modeled lipid-like entities (green and pink)" and "Unmodeled lipid-like density is shown pink" cannot be right. In Figure 4C, both green and pink densities appeared to be modeled.

Thank you. All these minor suggestions are corrected.